# Dynamic Trace Estimation

**Prathamesh Dharangutte**
Dept.of Computer Science & Engineering
New York University
ptd244@nyu.edu

**Christopher Musco**
Dept.of Computer Science & Engineering
New York University
cmusco@nyu.edu

## Abstract

We study a *dynamic* version of the implicit trace estimation problem. Given access to an oracle for computing matrix-vector multiplications with a dynamically changing matrix $A$, our goal is to maintain an accurate approximation to $A$'s trace using as few multiplications as possible. We present a practical algorithm for solving this problem and prove that, in a natural setting, its complexity is quadratically better than the standard solution of repeatedly applying Hutchinson's stochastic trace estimator. We also provide an improved algorithm assuming slightly stronger assumptions on the dynamic matrix $A$. We support our theory with empirical results, showing significant computational improvements on three applications in machine learning and network science: tracking moments of the Hessian spectral density during neural network optimization, counting triangles, and estimating natural connectivity in a dynamically changing graph.

## 1   Introduction

Implicit or "matrix-free" trace estimation is a ubiquitous computational primitive in linear algebra, which has become increasingly important in machine learning and data science. Given access to an oracle for computing matrix-vector products $Ax_1, \ldots, Ax_m$ between an $n \times n$ matrix $A$ and chosen vectors $x_1, \ldots, x_m$, the goal is to compute an approximation to $A$'s trace, $\mathrm{tr}(A) = \sum_{i=1}^{n} A_{ii}$. This problem arises when $A$'s diagonal entries cannot be accessed explicitly, usually because forming $A$ is computationally prohibitive. As an example, consider $A$ which is the Hessian matrix of a loss function involving a neural network. While forming the Hessian is infeasible when the network is large, backpropagation can be used to efficiently compute Hessian-vector products [32].

In other applications, $A$ is a *matrix function* of another matrix $B$. For example, if $B$ is a graph adjacency matrix, $\mathrm{tr}(B^3)$ equals six times the number of triangle in the graph [2]. Computing $A = B^3$ explicitly to evaluate the trace would require $O(n^3)$ time, while the matrix-vector multiplication $Ax = B \cdot (B \cdot (Bx))$ only requires $O(n^2)$ time. Similarly, in log-determinant approximation, useful in e.g. Bayesian log likelihood computation or determinantal point process (DPP) methods, we want to approximate the trace of $A = \log(B)$ [5, 17, 36]. Again, $A$ takes $O(n^3)$ time to form explicitly, but $Ax = \log(B)x$ can be computed in roughly $O(n^2)$ time using iterative methods like the Lanczos algorithm [21]. Dynamic versions of the log-determinant estimation problem have been studied due to applications in greedy methods for DPP inference [15].

In data science and machine learning, other applications of implicit trace estimation include matrix norm and spectral sum estimation [16, 41, 31], as well as methods for eigenvalue counting [9] and spectral density estimation [45, 27]. Spectral density estimation methods typically use implicit trace estimation to estimate *moments* of a matrix's eigenvalue distribution – i.e., $\mathrm{tr}(A), \mathrm{tr}(A^2), \mathrm{tr}(A^3)$, etc. – which can then be used to compute an approximation to that entire distribution. In deep learning, spectral density estimation is used to quickly analyze the spectra of weight matrices [33, 28] or to probe information about the Hessian matrix during optimization [13, 49]. Trace estimation has also been used for neural networks weight quantization [10, 34] and to understand training dynamics [44].

35th Conference on Neural Information Processing Systems (NeurIPS 2021).

## 1.1 Static Trace Estimation

The mostly widely used implicit trace estimation algorithm is Hutchinson's estimator [14, 22]. Letting $g_1, \ldots, g_\ell \in \mathbb{R}^n$ be random vectors with i.i.d. mean 0 and variance 1 entries (e.g., standard Gaussian or $\pm 1$ Rademachers), Hutchinson's approximates $\mathrm{tr}(A)$ via the average $h_\ell(A) = \frac{1}{\ell} \sum_{i=1}^{1} g_i^T (Ag_i)$. This estimator requires $\ell$ matrix-vector multiplications to compute. Its variance can be shown to be $O(\|A\|_F^2/\ell)$ and with high probability, when $\ell = O(1/\epsilon^2)$, we have the error guarantee [3, 30]:

$$|h_\ell(A) - \mathrm{tr}(A)| < \epsilon \|A\|_F. \tag{1}$$

While improvements on Hutchinson's estimator have been studied for restricted classes of matrices (positive semidefinite, sparse, nearly low-rank, etc.) [30, 40, 38, 36, 19], the method is the best known for general matrices – no techniques achieve guarantee (1) with $o(1/\epsilon^2)$ matrix-vector products.

## 1.2 Dynamic Trace Estimation

We explore a natural and widely applicable dynamic version of the implicit trace estimation problem: given access to a matrix-vector multiplication oracle for a *dynamically changing* matrix $A$, maintain an approximation to $A$'s trace. This problem arises in applications involving optimization in machine learning where we need to estimate the trace of a constantly changing Hessian matrix $H$ (or some function of $H$) during model training. In other applications, $A$ is dynamic because it is repeatedly modified by some algorithmic process. E.g., in the transit planning method of [43], edges are added to a network to optimally increase the "Estrada index" [11]. Evaluating this connectivity measure requires computing $\mathrm{tr}(\exp(B))$, where $B$ is the dynamically changing network adjacency matrix and $\exp(B)$ is a matrix exponential. A naive solution to the dynamic problem is to simply apply Hutchinson's estimator to every snapshot of $A$ as it changes over time. To achieve a guarantee like (1) for $m$ time steps, we require $O(m/\epsilon^2)$ matrix-vector multiplies. The goal of this paper is to improve on this bound when the changes to $A$ are *bounded*. Formally, we abstract the problem as follows:

---

**Problem 1** (Dynamic trace estimation)**.** *Let $A_1, ..., A_m$ be $n \times n$ matrices satisfying:*

    1.  $\|A_i\|_F \leq 1$, *for all* $i \in [1, m]$.      2.  $\|A_{i+1} - A_i\|_F \leq \alpha$, *for all* $i \in [1, m-1]$.

*Given implicit matrix-vector multiplication access to each $A_i$ in sequence, the goal is to compute trace approximations $t_1, \ldots, t_m$ for $\mathrm{tr}(A_1), ...., \mathrm{tr}(A_m)$ such that, for each $i \in 1, \ldots, m$,*

$$\mathbb{P}[|t_i - \mathrm{tr}(A_i)| \geq \epsilon] \leq \delta. \tag{2}$$

---

Above $A_1, \ldots, A_m$ represent different snapshots of a dynamic matrix at $m$ time steps. We require $\|A_i\|_F \leq 1$ only to simplify the form of our error bounds – no explicit rescaling is necessary for matrices with larger norm. If we assume $\|A_i\|_F \leq U$ for some (unknown) upper bound $U$, the guarantee of (2) would simply change to involve a $\epsilon U$ terms instead of $\epsilon$. The second condition bounds how much the matrices change over time. Again for simplicity, we assume a fixed upper bound $\alpha$ on the difference at each time step, but the algorithms presented in this paper will be adaptive to changing gaps between $A_i$ and $A_{i+1}$, and will perform better when these gaps are small on average. By triangle inequality, $\alpha \leq 2$, but in applications we typically have $\alpha \ll 1$, meaning that the changes in the dynamic matrix are small relative to its Frobenius norm. If this is not the case, there is no hope to improve on the naive method of applying Hutchinson's estimator repeatedly to each $A_i$.

**Note on Matrix Functions.** In many applications $A_i = f(B_i)$ for a dynamically changing matrix $B$. While we may have $\|A_{i+1} - A_i\|_F = \|f(B_{i+1}) - f(B_i)\|_F \gg \|B_{i+1} - B_i\|_F$ for functions like the matrix exponential, this is not an immediate issue. To improve on Hutchinson's estimator, the important requirement is simply that $\|A_{i+1} - A_i\|_F$ is small *in comparison* to $\|A_{i+1}\|_F$. As discussed in Section 5, this is typically the case for application involving matrix functions.

We will measure the complexity of any algorithm for solving Problem 1 in the **matrix-vector multiplication oracle model of computation**, meaning that we consider the cost of matrix-vector products (which are the only way $A_1, \ldots, A_m$ can be accessed) to be significantly larger than other computational costs. We thus seek solely to minimize the number of such products used [39]. The matrix-vector oracle model has seen growing interest in recent years as it generalizes both the matrix sketching and Krylov subspace models in linear algebra, naturally captures the true computational cost of algorithms in these classes, and is amenable to proving strong lower-bounds [37, 6].

## 1.3 Main Result

Our main result is an algorithm for solving Problem 1 more efficiently than Hutchinson's estimator:

**Theorem 1.1.** *For any $\epsilon, \delta, \alpha \in (0,1)$, the DeltaShift algorithm (Algorithm 1) solves Problem 1 with*

$$O\left(m \cdot \frac{\alpha \log(1/\delta)}{\epsilon^2} + \frac{\log(1/\delta)}{\epsilon^2}\right)$$

*total matrix-vector multiplications involving $A_1, \ldots, A_m$.*

For large $m$, the first term dominates the complexity in Theorem 1.1. For comparison, a tight analysis of Hutchinson's estimator [29] establishes that the naive approach requires $O\left(m \cdot \log(1/\delta)/\epsilon^2\right)$, which is worse than Theorem 1.1 by a factor of $\alpha$. A natural setting is when $\alpha = O(\epsilon)$, in which case Algorithm 1 requires $O(\log(1/\delta)/\epsilon)$ matrix-multiplications on average over $m$ time steps, in comparison to $O(\log(1/\delta)/\epsilon^2)$ for Hutchinson's estimator, a quadratic improvement in $\epsilon$.

To prove Theorem 1.1, we introduce a dynamic *variance reduction* scheme. By linearity of trace, $\mathrm{tr}(A_{i+1}) = \mathrm{tr}(A_i) + \mathrm{tr}(\Delta_i)$, where $\Delta_i = A_{i+1} - A_i$. Instead of directly estimating $\mathrm{tr}(A_{i+1})$, we combine previous estimate for $\mathrm{tr}(A_i)$ with an estimate for $\mathrm{tr}(\Delta_i)$, computed via Hutchinson's estimator. Each sample for Hutchinson's estimator applied to $\Delta_i$ requires just two matrix-vector multiplies: one with $A_i$ and one with $A_{i+1}$. At the same time, when $\Delta_i$ has small Frobenius norm (bounded by $\alpha$), we can estimate its trace more accurately than $\mathrm{tr}(A_{i+1})$[1]. While intuitive, this approach requires care to make work. In a naive implementation, error in estimating $\mathrm{tr}(\Delta_1), \mathrm{tr}(\Delta_2), \ldots$, compounds over time, eliminating any computational savings. To avoid this issue, we introduce a novel damping strategy that actually estimates $\mathrm{tr}(A_{i+1} - (1-\gamma)A_i)$ for a positive damping factor $\gamma$.

We compliment our main result with a nearly matching *conditional lower bound*: in Section 4 we argue that our DeltaShift method cannot be improved in the dynamic setting unless Hutchinson's estimator can be improved in the static setting. We also present an improvement to DeltaShift under more stringent, but commonly present, bounds on $A_1, \ldots, A_m$ and each $A_{i+1} - A_i$ than Problem 1.

## 1.4 Related Work

Prior work on implicit trace estimation and applications in machine learning is discussed in the beginning of this section. While there are no other methods that improve on Hutchinson's estimator in the dynamic setting, the idea of variance reduction has found applications in other work on implicit trace estimation [1, 12, 26, 30]. In these results, the trace of a matrix $A$ is estimated by decomposing $A = B + \Delta$ where $B$ has an easily computed trace (e.g., because it is low-rank) and $\|\Delta\|_F \ll \|A\|_F$, so $\mathrm{tr}(\Delta)$ is more easily approximated with Hutchinson's estimator than $\mathrm{tr}(A)$ directly.

## 2 Preliminaries

**Notation.** We let $B \in \mathbb{R}^{m \times k}$ denote a real-valued matrix with $m$ rows and $k$ columns. $x \in \mathbb{R}^n$ denotes a real-valued vector with $n$ entries. Subscripts like $B_i$ or $x_j$ typically denote a matrix or vector in a sequence, but we use double subscripts with matrices to denote entries: $B_{ij}$ being the entry at the $i^{\mathrm{th}}$ row and $j^{\mathrm{th}}$ column. Let $\sigma_\ell(B)$ denote the $\ell^{\mathrm{th}}$ singular value of $B$. $\|B\|_F$ denotes the Frobenius norm of $B$, $\sqrt{\sum_{i,j} B_{ij}^2} = \sum_\ell \sigma_\ell(B)^2$. $\|B\|_*$ denotes the nuclear norm, $\sum_\ell \sigma_\ell(B)$. We let $\mathbb{E}[v]$ and $\mathrm{Var}[v]$ denote the expectation and variance of a random variable $v$.

**Hutchinson's Estimator.** Our algorithm uses Hutchinson's trace estimator with Rademacher $\pm 1$ random variables as a subroutine. Specifically, let $g_1, \ldots, g_\ell \in \mathbb{R}^n$ be independent random vectors, with each entry $+1$ or $-1$ with probability $1/2$. Let $A \in \mathbb{R}^{n \times n}$. Hutchinson's estimator for $\mathrm{tr}(A)$ is:

$$h_\ell(A) = \frac{1}{\ell} \sum_{i=1}^{\ell} g_i^T (A g_i) \tag{3}$$

---

[1]While we consider the general, unstructured problem setting, we note that, if $A_i$ has additional structure, it is not necessarily easier to estimate the trace of $\Delta_i$ than that of $A_i$. For example, if $A_i$ is a PSD matrix then specialized trace estimation algorithms that improve on Hutchinson's method can be used [30]. Understanding dynamic traces estimation methods for sequences of structured matrices is a natural direction for future work.

**Fact 2.1** (Hutchinson's expectation and variance). *For any positive integer $\ell$ and matrix $A$ we have:*

$$\mathbb{E}[h_\ell(A)] = \text{tr}(A), \qquad \text{Var}[h_\ell(A)] = \frac{2}{\ell}\left(\|A\|_F^2 - \sum_{i=1}^n A_{ii}^2\right) \leq \frac{2}{\ell}\|A\|_F^2.$$

Fact 2.1 follows from simple calculations, found e.g. in [3]. Similar bounds hold when Hutchinson's estimator is implemented with different random variables. For example, random Gaussians also lead to a variance bound of $\frac{2}{\ell}\|A\|_F^2$. However, Rademachers tend to work better empirically. Given Fact 2.1, Chebyshev's inequality immediately implies a concentration bound for Hutchinson's estimator.

**Fact 2.2** (Chebyshev's Inequality). *For a random variable $X$ with mean $\mathbb{E}[X] = \mu$ and variance $\text{Var}[X] = \sigma^2$, for any $k \geq 1$, $\mathbb{P}(|X - \mu| \geq k\sigma) \leq 1/k^2$.*

**Claim 2.3.** *For any $\epsilon, \delta \in (0,1)$, if $\ell = \frac{2}{\epsilon^2\delta}$ then $\Pr\left[|h_\ell(A) - \text{tr}(A)| \geq \epsilon\|A\|_F\right] \leq \delta$.*

The $\delta$ dependence in Claim 2.3 can be improved from $\frac{1}{\delta}$ to $\log(1/\delta)$ via the Hanson-Wright inequality, which shows that $h_\ell(A)$ is a sub-exponential random variable [30, 35]. We also require Hanson-Wright to obtain our bound involving $\log(1/\delta)$. From this tighter result, Hutchinson's yields a total matrix-vector multiplication bound of $O(m \cdot \log(1/\delta)/\epsilon^2)$ for solving Problem 1 by simply applying the estimator in sequence to $A_1, \ldots, A_m$.

## 3  Main Algorithmic Result

As discussed in Section 1.2, a natural idea for solving Problem 1 with fewer than $O(m/\epsilon^2)$ queries is to take advantage of the small differences between $A_{i+1}$ and $A_i$ to compute a running estimate of the trace. In particular, instead of estimating $\text{tr}(A_1), \text{tr}(A_2), \ldots, \text{tr}(A_m)$ individually using Hutchinson's estimator, we denote $\Delta_i = A_i - A_{i-1}$ and use linearity of the trace to write:

$$\text{tr}(A_j) = \text{tr}(A_1) + \sum_{i=2}^j \text{tr}(\Delta_i). \tag{4}$$

By choosing a large $\ell_0$, we can compute an accurate approximation $h_{\ell_0}(A_1)$ to $\text{tr}(A_1)$. Then, for $j > 1$ and $\ell \ll \ell_0$, we can approximate $\text{tr}(A_j)$ via the following unbiased estimator:

$$\text{tr}(A_j) \approx h_{\ell_0}(A_1) + \sum_{i=2}^j h_\ell(\Delta_i) \tag{5}$$

Since $\|\Delta_i\|_F \leq \alpha \ll \|A_{i+1}\|_F$, we expect to approximate $\text{tr}(\Delta_2), \ldots, \text{tr}(\Delta_m)$ much more accurately than $\text{tr}(A_2), \ldots, \text{tr}(A_m)$ directly. At the same time, the estimator in (5) only incurs a 2 factor overhead in matrix-vector multiplies in comparisons to Hutchinson's: it requires $2 \cdot (m-1)\ell$ to compute $h_\ell(\Delta_2), \ldots, h_\ell(\Delta_m)$ versus $(m-1)\ell$ to compute $h_\ell(A_2), \ldots, h_\ell(A_m)$. The cost of the initial estimate $h_{\ell_0}(A_1)$ is necessarily higher, but can be amortized over time.

### 3.1  Our Approach

While intuitive, the problem with the approach above is that error compounds due to the sum in (5). Each $h_\ell(\Delta_i)$ is roughly $\alpha/\sqrt{\ell}$ away from $\text{tr}(\Delta_i)$, so after $j$ steps we naively expect total error $O(j \cdot \alpha/\sqrt{\ell})$. We can do slightly better by arguing that, due to their random nature, error actually accumulates as $O(\sqrt{j} \cdot \alpha/\sqrt{\ell})$, but regardless, there is accumulation. One option is to "restart" the estimation process: after some number of steps $q$, throw out all previous trace approximations, compute an accurate estimate for $\text{tr}(A_q)$, and for $j > q$ construct an estimator based on $\text{tr}(A_j) = \text{tr}(A_q) + \sum_{i=q+1}^{j-1} \text{tr}(\Delta_i)$. While possible to analyze theoretically, this approach turns out to be difficult to implement in practice due to several competing parameters (see details in Section 5).

Instead, we introduce a more effective approach based on a *damped variance reduction* strategy, which is detailed in Algorithm 1, which we call DeltaShift. Instead of being based on (4), DeltaShift uses the following recursive identity involving a fixed parameter $0 \leq \gamma < 1$ (to be chosen later):

$$\text{tr}(A_j) = (1-\gamma)\text{tr}(A_{j-1}) + \text{tr}(\widehat{\Delta}_j), \quad \text{where } \widehat{\Delta}_j = A_j - (1-\gamma)A_{j-1}. \tag{6}$$

---

**Algorithm 1** DeltaShift

---

**Input**: Implicit matrix-vector multiplication access to $A_1, ..., A_m \in \mathbb{R}^{n \times n}$, positive integers $\ell_0, \ell$, damping factor $\gamma \in [0, 1]$.

**Output**: $t_1, \dots, t_m$ approximating $\text{tr}(A_1), \dots, \text{tr}(A_m)$.

Initialize $t_1 \leftarrow \frac{1}{\ell_0} \sum_{i=1}^{\ell_0} g_i^T A_1 g_i$, where $g_1, \dots, g_{\ell_0} \in \mathbb{R}^n$ are random $\pm 1$ vectors

**for** $j \leftarrow 2$ **to** $m$ **do**

    Draw $\ell$ random $\pm 1$ vectors $g_1, \dots, g_\ell \in \mathbb{R}^n$

    $z_1 \leftarrow A_{j-1} g_1, \dots, z_\ell \leftarrow A_{j-1} g_\ell, \quad w_1 \leftarrow A_j g_1, \dots, w_\ell \leftarrow A_j g_\ell$

    $t_j \leftarrow (1 - \gamma) t_{j-1} + \frac{1}{\ell} \sum_{i=1}^{\ell} g_i^T (w_i - (1 - \gamma) z_i)$

**end for**

---

Given an estimate $t_{j-1}$ for $\text{tr}(A_{j-1})$, DeltaShift estimates $\text{tr}(A_j)$ by $(1 - \gamma) t_{j-1} + h_\ell(\widehat{\Delta}_j)$. This approach has several useful properties: 1) if $t_{j-1}$ is an unbiased estimate for $\text{tr}(A_{j-1})$, $t_j$ is an unbiased estimate for $\text{tr}(A_j)$, 2) $\|\widehat{\Delta}_j\|_F$ is not much larger than $\|\Delta_j\|_F$ if $\gamma$ is small, and 3) by shrinking $t_{j-1}$ by a factor of $(1 - \gamma)$ when computing $t_j$, we reduces the variance of this leading term. The last property ensures that error does not accumulate over time, leading to our main result:

**Theorem 1.1** (Restated). *For any $\epsilon, \delta, \alpha \in (0, 1)$, Algorithm 1 run with $\gamma = \alpha$, $\ell_0 = O\left(\log(1/\delta)/\epsilon^2\right)$, and $\ell = O\left(\alpha \log(1/\delta)/\epsilon^2\right)$ solves Problem 1. In total, it requires*

$$O\left(m \cdot \frac{\alpha \log(1/\delta)}{\epsilon^2} + \frac{\log(1/\delta)}{\epsilon^2}\right)$$

*matrix-vector multiplications with $A_1, \dots, A_m$.*

The full proof of Theorem 1.1 relies on the Hanson-Wright inequality, and is given in Appendix B. Here, we give a simple proof of essentially the same statement, but with a slightly weaker dependence on the failure probability $\delta$.

*Proof.* Let $\gamma = \alpha$, $\ell_0 = \frac{2}{\epsilon^2 \delta}$, and $\ell = \frac{8\alpha}{\epsilon^2 \delta}$. The proof is based on an inductive analysis of the variance of $t_j$, the algorithms estimate for $\text{tr}(A_j)$. Specifically, we claim that that for $j = 1, \dots, m$:

$$\text{Var}[t_j] \le \delta \epsilon^2. \tag{7}$$

For the base case, $j = 1$, (7) follows directly from Fact 2.1 because $t_1$ is simply Hutchinson's estimator applied to $A_1$, and $\|A_1\|_F \le 1$. For the inductive case, $t_j$ is the sum of two independent estimators, $t_{j-1}$ and $h_\ell(\widehat{\Delta}_j)$. So, to bound its variance, we just need to bound the variance of these two terms. To address the second, note that by triangle inequality, $\|\widehat{\Delta}_j\|_F = \|A_j - (1-\gamma)A_{j-1}\|_F \le \|A_j - A_{j-1}\|_F + \gamma\|A_{j-1}\|_F \le 2\alpha$. Thus, by Fact 2.1, $\text{Var}[h_\ell(\widehat{\Delta}_j)] \le \frac{8}{\ell}\alpha^2$. Combined with the inductive assumption that $\text{Var}[t_{j-1}] \le \delta\epsilon^2$, we have:

$$\text{Var}[t_j] = (1 - \gamma)^2 \text{Var}[t_{j-1}] + \text{Var}[h_\ell(\widehat{\Delta}_j)] \le (1 - \alpha)^2 \delta\epsilon^2 + \frac{8\alpha^2}{\ell} \le (1 - \alpha)\delta\epsilon^2 + \alpha\delta\epsilon^2 = \delta\epsilon^2.$$

This proves (7), and by Chebyshev's inequality we thus have $\Pr\left[|t_j - \text{tr}(A_j)| \ge \epsilon\right] \le \delta$ for all $j$. $\square$

### 3.2 Selecting $\gamma$ in Practice

While DeltaShift is simple to implement, in practice, its performance is sensitive to the choice of $\gamma$. For the Theorem 1.1 analysis, we assume $\gamma = \alpha$, but $\alpha$ may not be known apriori, and may change over time. To address this issue, we describe a way to select a *near optimal* $\gamma$ at each time step $j$ (the choice may vary over time) with very little additional computational overhead. Let $v_{j-1} = \text{Var}[t_{j-1}]$ be the variance of our estimator for $\text{tr}(A_{j-1})$. We have that $v_j = (1 - \gamma)^2 v_{j-1} + \text{Var}[h_\ell(A_j - (1 - \gamma)A_{j-1})] \le (1 - \gamma)^2 v_{j-1} + \frac{2}{\ell}\|A_j - (1 - \gamma)A_{j-1}\|_F^2$. At time step $j$, a natural goal is to choose damping parameter $\gamma^*$ that minimizes this upper bound on the variance of $t_j$:

$$\gamma^* = \arg\min_\gamma \left[(1 - \gamma)^2 v_{j-1} + \frac{2}{\ell}\|\widehat{\Delta}_j\|_F^2\right], \tag{8}$$

where $\widehat{\Delta}_j = A_j - (1-\gamma)A_{j-1}$ as before. While (8) cannot be computed directly, observing that $\|B\|_F^2 = \text{tr}(B^T B)$ for any matrix $B$, the above quantity can be estimated as $\tilde{v}_j = (1-\gamma)^2 \tilde{v}_{j-1} + \frac{2}{\ell} h_\ell(\widehat{\Delta}_j^T \widehat{\Delta}_j)$, where $\tilde{v}_{j-1}$ is an estimate for $v_{j-1}$. The estimate $h_\ell(\widehat{\Delta}_j^T \widehat{\Delta}_j)$ can be computed using exactly the same $\ell$ matrix-vector products with $A_j$ and $A_{j-1}$ that are used to estimate $\text{tr}(\widehat{\Delta}_j)$, so there is little computational overhead. Moreover, since $\widehat{\Delta}_j^T \widehat{\Delta}_j$ is positive semidefinite, as long as $\ell \geq \log(1/\delta)$, we will obtain a *relative error approximation* to its trace with probability $1 - \delta$ [3].

An alternative approach to estimating $v_j$ would be to simply compute the empirical variance of the average $h_\ell(\widehat{\Delta}_j)$, but this requires fixing $\gamma$. An advantage of our closed form approximation is that it can be used to analytically optimize $\gamma$. Specifically, expanding $\widehat{\Delta}_j^T \widehat{\Delta}_j$, we have that:

$$\tilde{v}_j = (1-\gamma)^2 \tilde{v}_{j-1} + \frac{2}{\ell} \left( h_\ell(A_j^T A_j) + (1-\gamma)^2 h_\ell(A_j^T A_j) - 2(1-\gamma)h_\ell(A_{j-1}^T A_j) \right). \quad (9)$$

Above, each estimate $h_\ell$ is understood to use the same set of random vectors. Taking the derivative and setting to zero, we have that the minimizer of (9), denoted $\tilde{\gamma}^*$, equals:

$$\tilde{\gamma}^* = 1 - \frac{2h_\ell(A_{j-1}^T A_j)}{\ell \tilde{v}_{j-1} + 2h_\ell(A_{j-1}^T A_{j-1})}. \quad (10)$$

This formula for $\tilde{\gamma}^*$ motivates an essentially parameter free version of DeltaShift, which is used in our experimental evaluation (Algorithm 2 in Appendix A). The only input to the algorithm is the number of matrix-vector multiplies used at each time step, $\ell$. For simplicity, unlike Algorithm 1, we do not use a larger number of matrix-vector multiplies when estimating $A_1$. This leads to somewhat higher error for the first matrices in the sequence $A_1, \ldots, A_m$, but error quickly falls for large $j$.

## 4 Algorithm Improvements and Lower Bound

In this section, we prove a lower bound showing that, in general, Theorem 1.1 is likely optimal. On the other hand, we show that, if we make a slightly stronger assumption on $A_1, \ldots, A_m$ and the dynamic updates $\Delta_i = A_{i+1} - A_i$, an improvement on DeltaShift is possible.

### 4.1 Lower Bound

As noted, for a large number of time steps $m$, the matrix-vector multiplication complexity of DeltaShift is dominated by the leading term in Theorem 1.1, $O(m \cdot \alpha \log(1/\delta)/\epsilon^2)$. We show that it is unlikely an improvement on this term can be obtained in general:

**Lemma 4.1.** *Suppose there is an algorithm $\mathcal{S}$ that solves Prob. 1 with $o(m \cdot \alpha \log(1/\delta)/\epsilon^2)$ total matrix-vector multiplies with $A_2, \ldots, A_m$, and* any number *of matrix-vector multiplies with $A_1$ when $\alpha = 1/(m-1)$. Then there is an algorithm $\mathcal{T}$ that achieves (1) for a single $A$ with $o(\log(1/\delta)/\epsilon^2)$ matrix-vector multiplies.*

*Proof.* The proof is via a direct reduction. Given a matrix $A$, positive integer $m > 1$, and parameter $\alpha = \frac{1}{m-1}$, construct the sequence of matrices:

$$A_1 = 0, \qquad A_2 = \alpha \cdot A, \qquad \ldots \qquad A_{1/\alpha} = (1-\alpha)A, \qquad A_m = A$$

Since $A = 0$, and every $A_2, \ldots, A_m$ is a scaling of $A$, any algorithm $\mathcal{S}$ satisfying the assumption of Lemma 4.1 can be implemented with $o\left((m-1) \cdot \alpha \log(1/\delta)/\epsilon^2\right) = o(\log(1/\delta)/\epsilon^2)$ matrix-vector multiplications with $A$. Moreover, if $\mathcal{S}$ is run on this sequence of matrices, on the last step it outputs an approximation $t_m$ to $\text{tr}(A)$ with $\Pr[|t_m - \text{tr}(A)| \geq \epsilon] \leq \delta$. So algorithm $\mathcal{T}$ can simply simulate $\mathcal{S}$ on $A_1, \ldots, A_m$ and return its final estimate to satisfy (1). $\qquad\square$

Lemma 4.1 is a *conditional* lower-bound on matrix-vector query algorithms for solving Problem 1: if Hutchinson's estimator cannot be improved for static trace estimation (and it hasn't been for 30 years) then DeltaShift cannot be improved for dynamic trace estimation. We believe the bound could be made *unconditional* through a slight generalization of existing lower bounds on trace estimation in the matrix-vector multiplication model [30, 46].

## 4.2 Improved Algorithm

A recent improvement on Hutchinson's estimator, called Hutch++, was described in [30]. For the *static* trace estimation problem, Hutch++ achieves a matrix-vector multiplication complexity of $O(1/\epsilon)$ to compute a relative error $(1 \pm \epsilon)$ approximation to the trace of any positive semi-definite matrix (PSD), improving on the $O(1/\epsilon^2)$ required by Hutchinson's. It does so via a variance reduction method (also used e.g. in [12]) which allocates some matrix-vector products to a randomized SVD algorithm which approximates the top singular vector subspace of $A$. This approximate subspace is projected off of $A$ and Hutchinson's used to estimate the trace of the remainder.

In our setting it is not realistic to assume PSD matrices – while in many applications $A_1, \ldots, A_m$ are all PSD, it is rarely the case the $\Delta_1, \ldots, \Delta_{m-1}$ are. Nevertheless, we can take advantage of a more general bound proven in [30] for any matrix:

**Fact 4.2** (Hutch++ expectation and variance). *Let $h_\ell^{++}(A)$ be the Hutch++ estimator of [30] applied to any matrix $A$ with $\ell$ matrix-vector multiplications. We have:*

$$\mathbb{E}[h_\ell^{++}(A)] = \operatorname{tr}(A), \qquad\qquad \operatorname{Var}[h_\ell^{++}(A)] \leq \frac{16}{\ell^2} \|A\|_*^2.$$

Recall that $\|A\|_*$ denotes the nuclear norm of $A$. Comparing to the variance of Hutchinson's estimator from Fact 2.1, notice that the variance of Hutch++ depends on $\frac{1}{\ell^2}$ instead of $\frac{1}{\ell}$, implying faster convergence as the number of matrix-vector products, $\ell$, increases. A trade-off is that the variance scales with $\|A\|_*^2$ instead of $\|A\|_F^2$. $\|A\|_*^2$ is strictly larger, and possible a factor of $n$ larger than $\|A\|_F^2$. However, for matrices that are rank $k$, $\|A\|_*^2 \leq k\|A\|_F^2$, so the norms are typically much closer for **low-rank or nearly low-rank matrices**. In many problems, $\Delta_1, \ldots, \Delta_m$ may have low-rank structure, in which case, an alternative based on Hutch++ provides better performance.

Formally, we introduce a new variant of Problem 1 to capture this potential improvement.

---

**Problem 2** (Dynamic trace estimation w/ Nuclear norm assumption). *Let $A_1, ..., A_m$ satisfy:*

1. $\|A_i\|_* \leq 1$, *for all $i \in [1, m]$.*    2. $\|A_{i+1} - A_i\|_* \leq \alpha$, *for all $i \in [1, m-1]$.*

*Given matrix-vector multiplication access to each $A_i$ in sequence, the goal is to compute trace approximations $t_1, \ldots, t_m$ for $\operatorname{tr}(A_1), \ldots, \operatorname{tr}(A_m)$ such that, for all $i$, $\mathbb{P}[|t_i - \operatorname{tr}(A_i)| \geq \epsilon] \leq \delta$.*

---

In Appendix C we prove the following result on a variant of DeltaShift that we call DeltaShift++:

**Theorem 4.3.** *For any $\epsilon, \delta, \alpha \in (0, 1)$, DeltaShift++ (Algorithm 4) solves Problem 2 with*

$$O\left(m \cdot \frac{\sqrt{\alpha/\delta}}{\epsilon} + \frac{\sqrt{1/\delta}}{\epsilon}\right)$$

*total matrix-vector multiplications involving $A_1, \ldots, A_m$.*

Theorem 4.3 is stronger than Theorem 1.1 for vanilla DeltaShift in that it has a linear instead of a quadratic dependence on $\epsilon$. In particular, its leading term scales as $\sqrt{\alpha/\epsilon^2}$, whereas Theorem 1.1 scaled with $\alpha/\epsilon^2$. However, the result does require stronger assumptions on $A_1, \ldots, A_m$ and each $\Delta_i = A_{i+1} - A_i$ in that Problem 2 requires these matrices to have bounded nuclear norm instead of Frobenius norm. Since the nuclear norm of a matrix is strictly larger than its Frobenius norm, these requirements are stronger than those of Problem 1. As we will show in Section 5, the benefit of improved $\epsilon$ dependence often outweights the cost of these more stringent assumptions.

## 5 Experiments

We show that our proposed algorithm outperforms three alternatives on both synthetic and real-world trace estimation problems. Specifically, we evaluate the following methods:

**Hutchinson.** The naive method of estimating each $\operatorname{tr}(A_1), \ldots, \operatorname{tr}(A_m)$ using an independent Hutchinson's estimator, as discussed in Section 2.

**NoRestart.** The estimator of (5), which uses the same variance reduction strategy as DeltaShift for all $j \geq 2$, but does not restart or add damping to reduce error accumulation.

**Restart.** The estimator discussed in Sec. 3.1, which periodically restarts the variance reduction strategy, using Hutchinson's to obtain a fresh estimate for $\text{tr}(A_j)$. Pseudocode is in Appendix A.

**DeltaShift.** Our parameter free, damped variance reduction estimator detailed in Appendix A.

We allocated a fixed number of matrix-vector queries, $Q$, to be used over all time steps $1, \ldots, m$. For Hutchinson and DeltaShift, the same number of vectors $Q/m$ was used at each step. For Restart and NoRestart, the distribution was non-uniform, and parameter selections are described in Appendix D.

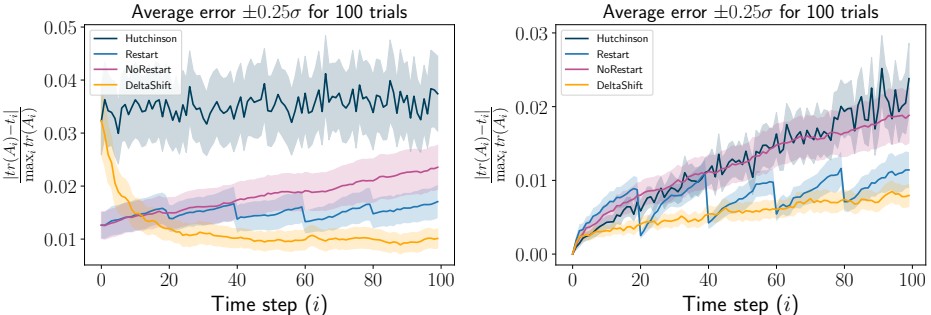

(a) Synthetic data with low perturbation      (b) Synthetic data with significant perturbation

Figure 1: Comparison of DeltaShift with Hutchinson and NoRestart on synthetic data with $Q = 10^4$.

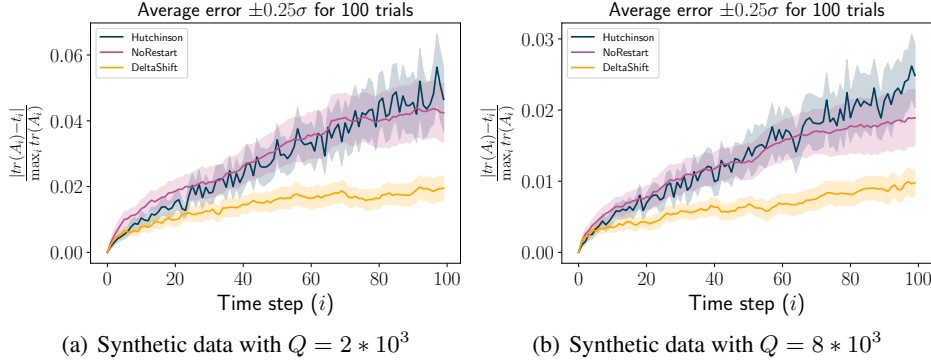

(a) Synthetic data with $Q = 2 * 10^3$      (b) Synthetic data with $Q = 8 * 10^3$

Figure 2: Comparison of DeltaShift with Hutchinson and NoRestart on synthetic data.

**Synthetic data:** To simulate the dynamic setting, we generate a random matrix $A \in \mathbb{R}^{n \times n}$ and add random perturbations for each of 100 time steps. We consider two cases: low (Fig. 1(a)) and significant (Fig. 1(b)) perturbations, the exact details of which, as well as the allocation of matrix-vector products for NoRestart and Restart, are discussed in Appendix D. We report scaled absolute error between the estimator at time, $t_j$, and the true trace $\text{tr}(A_j)$. As expected, Hutchinson is outperformed even by NoRestart when perturbations are small. DeltaShift performs best, and its error actually improve slightly over time. DeltaShift also performs best for the large perturbation experiment. We note that choosing the multiple parameters for the Restart method was a challenge in comparison to DeltaShift. Tuning the method becomes infeasible for larger experiments, so we exclude this method in our other experiments. That includes for the plots in Fig. 2, which show that DeltaShift continues to outperform Hutchinson and NoRestart for lower values of $Q$.

**Counting triangles:** Our first real-data experiment is on counting triangles in a dynamic unweighted, undirected graph $G$ via the fact that the number of triangles equals $\frac{1}{6}\text{tr}(B^3)$, where $B$ is the adjacency matrix. The graph dataset we use is the Wikipedia vote network dataset with 7115 nodes [25, 24]. At each timestep we perturb the graph by adding a random $k$-clique, for $k$ chosen uniformly between 10 and 150. After 75 time steps, we start randomly deleting among the subgraphs added. We follow the same setup for number of matrix-vector products used by the estimators and the error reported as in the synthetic experiments(Appendix D). Note that for this particular application, the actual number of matrix-vector multiplications with $B$ is $3Q$, since each oracle call computes $B(B(Bx))$. As seen in Fig. 3, DeltaShift provides the best estimates overall.

**Estimation natural connectivity:** To evaluate the DeltaShift++ algorithm introduced in Section 4, we address an application in [43] on estimating natural connectivity in a dynamic graph, which is a function of $\text{tr}(\exp(B))$ for adjacency matrix $B$. This problem has also been explored in [7, 8, 4]. We

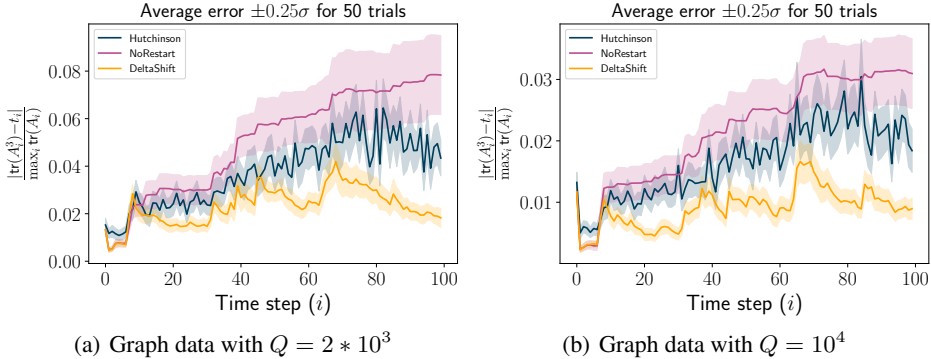

(a) Graph data with $Q = 2 * 10^3$       (b) Graph data with $Q = 10^4$

Figure 3: Comparison of DeltaShift with Hutchinson and NoRestart for triangle counting experiment.

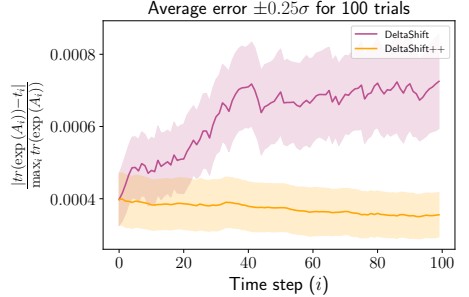

Figure 4: Error of DeltaShift and DeltaShift++ for dynamic estimation of natural connectivity.

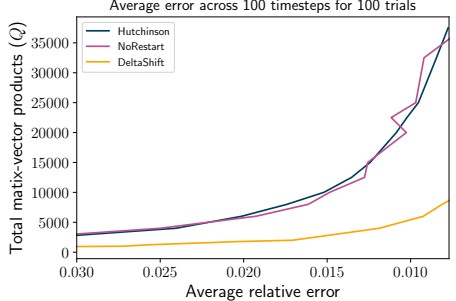

Figure 5: Average error vs. computational cost for synthetic data with large perturbations.

use the road network data Gleich/minnesota (available at `https://sparse.tamu.edu/Gleich/minnesota`). We perturb the graph over time by choosing two nodes at random and adding an edge between them, and use the Lanczos method to approximate matrix-vector products with $\exp(B)$. We find that DeltaShift++ performs better than DeltaShift (Fig. 4), as the change in $\exp(B)$ tends to be nearly low-rank, and thus have small nuclear norm (see [4] for details). Both DeltaShift and DeltaShift++ perform significantly better than naive Hutchinson's when 100 matrix-vector products are used per time step.

The key takeaway from the experiments above is that DeltaShift and DeltaShift++ are able to obtain good dynamic trace approximations in far fewer matrix-vector products compared to Hutchinson's and other methods, resulting in considerable computational savings. This is made evident in Figure 5, which plots average relative error across all time steps vs. total number of matrix-vector products($Q$), for various values of $Q$. In order to achieve the accuracy level as DeltaShift, Hutchinson's requires substantially more matrix-vector products.

**Hessian spectral density:** Finally, we evaluate the performance of DeltaShift on an application pertaining to a dynamically changing Hessian matrix, $H$, involved in training a neural network. As discussed in Section 1, a common goal is to approximate the *spectral density* of $H$. Most methods for doing so, like the popular Kernel Polynomial Method [45], require computing the trace of polynomials of the matrix $H$. We consider the sequence of Chebyshev polynomials $T_0, \ldots, T_q$, and estimate $\text{tr}(T_0(H)), \ldots, \text{tr}(T_q(H))$. Other polynomial basis sets can also be used (e.g., Legendre polynomials). Experimental details are discussed in section Appendix D, but we summarized the results here. We implement matrix vector products with $H$ using the PyHessian library [47], and report

Table 1: Average relative error for trace of Chebyshev polynomials of Hessian.

|          | Hutchinson | NoRestart | DeltaShift |
|----------|-----------|-----------|-----------|
| $T_1(H)$ | 2.5e-02   | 3.7e-02   | 1.7e-02   |
| $T_2(H)$ | 1.2e-06   | 1.7e-06   | 8.0e-07   |
| $T_3(H)$ | 4.0e-02   | 4.1e-02   | 3.1e-02   |
| $T_4(H)$ | 1.5e-06   | 1.7e-06   | 1.0e-06   |
| $T_5(H)$ | 2.1e-02   | 4.3e-02   | 1.9e-02   |

Table 2: Average relative error for trace of Chebyshev polynomials of Hessian.

|          | Hutchinson | NoRestart | DeltaShift |
|----------|-----------|-----------|-----------|
| $T_1(H)$ | 1.9e-02   | 5.0e-02   | 1.5e-02   |
| $T_2(H)$ | 1.2e-06   | 2.9e-06   | 9.9e-07   |
| $T_3(H)$ | 7.7e-02   | 9.4e-02   | 6.1e-02   |
| $T_4(H)$ | 1.7e-06   | 2.8e-06   | 1.5e-06   |
| $T_5(H)$ | 2.1e-02   | 4.2e-02   | 1.8e-02   |

average error over 25 training epochs for the Hessian of a ResNet model with 269722 parameters trained it on the CIFAR-10 dataset. As it is impossible to compute the true trace of these matrices, we use Hutchinson's estimator with a greater number of queries as placeholder for ground-truth, and compare the performance against the computed values. As can be seen in Tables 1 and 2, DeltaShift obtains uniformly better approximation to the trace values, although the improvement is small. This makes sense, as more progress on each training epoch implies a greater change in the Hessian over time, meaning $\alpha$ is larger and thus DeltaShift's advantage over Hutchinson's is smaller.

## Acknowledgements

We would like to think Cameron Musco for helpful discussions, as well as the paper referees for detailed feedback. This work was supported by NSF Award #2045590.

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
