# A Algorithms

Below we include detailed pseudocode for algorithms described in the main text.

---

**Algorithm 2** Parameter Free DeltaShift

---

**Input**: Implicit matrix-vector multiplication access to $A_1, ..., A_m \in \mathbb{R}^{n \times n}$, positive integer $\ell$.
**Output**: $t_1, \ldots, t_m$ approximating $\text{tr}(A_1), \ldots, \text{tr}(A_m)$.

    Draw $\ell$ random $\pm 1$ vectors $g_1, \ldots, g_\ell \in \mathbb{R}^n$
    $z_1 \leftarrow A_1 g_1, \ldots, z_\ell \leftarrow A_1 g_\ell$
    $N \leftarrow \frac{1}{\ell} \sum_{i=1}^{\ell} z_i^T z_i$                      (estimate for $\|A_1\|_F^2$)
    Initialize $t_1 \leftarrow \frac{1}{\ell} \sum_{i=1}^{\ell} g_i^T z_i$ and $v_1 \leftarrow \frac{2}{\ell} N$
    **for** $j \leftarrow 2$ **to** $m$ **do**
        Draw $\ell$ random $\pm 1$ vectors $g_1, \ldots, g_\ell \in \mathbb{R}^n$
        $z_1 \leftarrow A_{j-1} g_1, \ldots, z_\ell \leftarrow A_{j-1} g_\ell$, $w_1 \leftarrow A_j g_1, \ldots, w_\ell \leftarrow A_j g_\ell$
        $N \leftarrow \frac{1}{\ell} \sum_{i=1}^{\ell} z_i^T z_i$, $M \leftarrow \frac{1}{\ell} \sum_{i=1}^{\ell} w_i^T w_i$, $C \leftarrow \frac{1}{\ell} \sum_{i=1}^{\ell} w_i^T z_i$   (estimate for $\text{tr}(A_{j-1}^T A_{j-1})$,
                                                            $\text{tr}(A_j^T A_j) \& \text{tr}(A_{j-1}^T A_j))$
        $\gamma \leftarrow 1 - \frac{2C}{\ell \tilde{v}_{j-1} + 2N}$                                   (optimal damping factor)
        $t_j \leftarrow (1 - \gamma) t_{j-1} + \frac{1}{\ell} \sum_{i=1}^{\ell} g_i^T \left( w_i - (1 - \gamma) z_i \right)$
        $v_j \leftarrow (1 - \gamma)^2 v_{j-1} + \frac{2}{\ell} \left( N + (1 - \gamma)^2 M - 2(1 - \gamma) C \right)$
    **end for**

---

**Algorithm 3** Dynamic Trace Estimation w/ Restarts

---

**Input**: Implicit matrix-vector multiplication access to $A_1, ..., A_m \in \mathbb{R}^{n \times n}$, positive integers $\ell_0, \ell, q \leq m$.
**Output**: $t_1, \ldots, t_m$ approximating $\text{tr}(A_1), \ldots, \text{tr}(A_m)$.

    Draw $\ell_0$ random $\pm 1$ vectors $g_1, \ldots, g_{\ell_0} \in \mathbb{R}^n$
    Initialize $t_1 \leftarrow \frac{1}{\ell_0} \sum_{i=1}^{\ell_0} g_i^T A_1 g_i$
    **for** $j \leftarrow 2$ **to** $m$ **do**
        **if** $j \equiv 1 \pmod{n}$ **then**
            Draw $\ell_0$ random $\pm 1$ vectors $g_1, \ldots, g_{\ell_0} \in \mathbb{R}^n$
            $t_j \leftarrow \frac{1}{\ell_0} \sum_{i=1}^{\ell_0} g_i^T A_j g_i$
        **else**
            Draw $\ell_0$ random $\pm 1$ vectors $g_1, \ldots, g_\ell \in \mathbb{R}^n$
            $t_j \leftarrow t_{j-1} + \frac{1}{\ell} \sum_{i=1}^{\ell} g_i^T (A_j - A_{j-1}) g_i$
        **end if**
    **end for**

---

# B High Probability Proofs

In this section, we give a full proof of Theorem 1.1 with the correct logarithmic dependence on $1/\delta$. Before doing so, we collect several definitions and results required for proving the theorem.

**Definition 1.** *[42] A random variable* X *with* $\mathbb{E}[X] = \mu$ *is* sub-exponential *with parameters* $(\nu, \beta)$ *if its moment generating function satisfies:*

$$\mathbb{E}[e^{\lambda(X-\mu)}] \leq e^{\frac{\nu^2 \lambda^2}{2}} \quad \text{for all } |\lambda| < \frac{1}{\beta}.$$

**Claim B.1.** *[42] Any sub-exponential random variable with parameters* $(\nu, \beta)$ *satisfies the tail bound*

$$\Pr[|X - \mu| \geq t] \leq \begin{cases} 2e^{\frac{-t^2}{2\nu^2}} & \text{if } 0 \leq t \leq \frac{\nu^2}{\beta} \\ 2e^{\frac{-t}{2\beta}} & \text{for } t > \frac{\nu^2}{\beta}. \end{cases}$$

**Claim B.2.** *Let* $X_1, X_2, \ldots, X_k$ *be independent random variables with mean* $\mu_1, \ldots, \mu_k$ *and sub-exponential parameters* $(\nu_1, \beta_1), \ldots (\nu_k, \beta_k)$*, then* $\sum_{i=1}^{k} a_i X_i$ *is sub-exponential with parameters* $(\nu_*, \beta_*)$ *where,*

$$\nu_* = \sqrt{\sum_{i=1}^{k} a_i^2 \nu_i^2} \quad and \quad \beta_* = \max_{i=1,\ldots,k} a_i \beta_i$$

*Proof.* The proof is straight-forward by computing the moment generating function of $\sum_{i=1}^{k} a_i X_i$ using the independence of $X_1, X_2, \ldots, X_k$. Specifically, for $|\lambda| < 1/(\max_{i=1,\ldots,k} a_i \beta_i)$ we have:

$$\mathbb{E}[e^{\lambda \sum_{i=1}^{k} a_i(X_i - \mu_i)}] = \prod_{i=1}^{k} \mathbb{E}[e^{\lambda a_i(X_i - \mu_i)}]$$

$$\leq \prod_{i=1}^{k} e^{\frac{\lambda^2 a_i^2 \nu_i^2}{2}} = e^{\frac{\lambda^2 \nu_*^2}{2}}.$$

$\square$

As discussed, a tight analysis of Hutchinson's estimator, and also our DeltaShift algorithm, relies on the *Hanson-Wright inequality* [18], which shows that any quadratic form involving a vector with i.i.d. sub-Gaussian entries is a sub-exponential random variable. Thanks to existing concentration results for sub-exponential's this allows us to obtain a better dependence on the failure probability than given by the cruder Chebyshev's inequality presented in the paper's main text. Specifically, we use the following version of the inequality:

**Claim B.3.** *[Corollary of Theorem 1.1, [35]] For* $A \in \mathbb{R}^{n \times n}$*, let* $h_\ell(A)$ *be Hutchinson's estimator as defined in Section 2, implemented with Rademacher random vectors.* $h_\ell(A)$ *is a sub-exponential random variable with parameters*

$$\nu = \frac{c_1 \|A\|_F}{\sqrt{\ell}} \quad and \quad \beta = \frac{c_2 \|A\|_2}{\ell},$$

*where* $c_1, c_2$ *are absolute constants.*

*Proof.* Recall that $h_\ell(A)$ is an average of $\ell$ independent random variables, each of the form $g^T A g$, where $g \in \mathbb{R}^n$ is a vector with independent $\pm 1$ Rademacher random entries. We start by decoupling $g^T A g$ in two sums involving diagonal and off-diagonal terms in $A$:

$$g^T A g = \sum_{i=1}^{n} g_i^2 A_{ii} + \sum_{i,j:i \neq j}^{n} A_{ij} g_i g_j.$$

Here $g_i$ denotes the $i^{\text{th}}$ entry of $g$. Since each $g_i$ is sampled i.i.d. from a $\pm 1$ Rademacher distribution, the first term is constant with value $\sum_{i=1}^{n} A_{ii} = \operatorname{tr}(A)$. [35] derive a bound on the moment generating function for the off-diagonal term, which they denote $S = \sum_{i,j:i \neq j}^{n} A_{ij} g_i g_j$. Specifically, they show that

$$\mathbb{E}\left[e^{\lambda S}\right] \leq e^{c_1^2 \|A\|_F^2 \lambda^2 / 2}, \quad \text{for all } |\lambda| < \frac{1}{c_2 \|A\|_2},$$

where $c_1, c_2$ are positive constants. As $S$ is mean zero, we conclude that it is sub-exponential with parameters $(c_1\|A\|_F, c_2\|A\|_2)$ (refer to Definition 1), and thus $g^T A g$ (which is just $S$ added to a constant) is sub-exponential with same parameters. Finally, from Claim B.2, we immediately have that $h_\ell(A)$ is sub-exponential with parameters $\left(\frac{c_1\|A\|_F}{\sqrt{\ell}}, \frac{c_2\|A\|_2}{\ell}\right)$. Note that, while we only consider $\pm 1$ Rademacher random vectors, a similar analysis can be performed for any i.i.d. sub-Gaussian random entries by showing that the diagonal term is itself subexponential (it will no longer be constant). The result will involve additional constants depending on the choice of $g_i$. In the case when $g_i$ are i.i.d. standard normals, the diagonal term is a scaled chi-squared random variable. $\square$

Now, we are ready to move on to the main result.

**Theorem 1.1** (Restated). *For any $\epsilon, \delta, \alpha \in (0,1)$, Algorithm 1 run with $\gamma = \alpha$, $\ell_0 = O\left(\frac{\log(1/\delta)}{\epsilon^2}\right)$, and $\ell = O\left(\frac{\alpha \log(1/\delta)}{\epsilon^2}\right)$ solves Problem 1. In total, it requires*

$$O\left(m \cdot \frac{\alpha \log(1/\delta)}{\epsilon^2} + \frac{\log(1/\delta)}{\epsilon^2}\right)$$

*matrix-vector multiplications with $A_1, \ldots, A_m$.*

*Proof.* The proof is by induction. Let $t_1, \ldots, t_m$ be the estimators for $\operatorname{tr}(A_1), \ldots, \operatorname{tr}(A_m)$ returned by Algorithm 1. We claim that, for all $j = 1, \ldots, m$, $t_j$ is sub-exponential with parameters

$$\nu_j \leq \frac{\epsilon}{2\sqrt{\log(2/\delta)}}, \qquad\qquad \beta_j \leq \frac{\epsilon^2}{4\log(2/\delta)}. \qquad (11)$$

If we can prove (11), the theorem immediately follows by applying Claim B.1 with $t = \epsilon$ to the random variable $t_j$. Recall that $\mathbb{E}[t_j] = \operatorname{tr}(A_j)$

First consider the base case, $t_1 = h_{l_0}(A_1)$. By Claim B.3, $h_{l_0}(A_1)$ is sub-exponential with parameters $\left(\frac{c_1}{\sqrt{\ell_0}}, \frac{c_2\|A_1\|_2}{\ell_0}\right)$. Noting that $\|A_1\|_2 \leq \|A_1\|_F \leq 1$ and setting constants appropriately on $\ell_0$ gives the bound.

Next consider the inductive case. Recall that $t_j = (1-\gamma)t_{j-1} + h_\ell(\widehat{\Delta}_j)$, where $\widehat{\Delta}_j = A_j - (1-\gamma)A_{j-1}$. As shown in Section 3.1, $\|\widehat{\Delta}_j\|_F \leq 2\alpha$. So by Claim B.3, $h_\ell(\widehat{\Delta}_1)$ is sub-exponential with parameters $\left(\frac{2c_1\alpha}{\sqrt{\ell}}, \frac{2c_2\alpha}{\ell}\right)$. As long as $\ell = c \cdot \frac{\alpha \log(2/\delta)}{\epsilon^2}$ for sufficiently large constant $c$, we therefore have by Claim B.2 that

$$\beta_j = \max\left[(1-\gamma)\beta_{j-1}, \frac{2c_2\alpha}{\ell}\right] \leq \frac{\epsilon^2}{4\log(2/\delta)}.$$

Note that above we used the $\|\widehat{\Delta}_j\|_2 \leq \|\widehat{\Delta}_j\|_F \leq 2\alpha$. Setting $\gamma = \alpha$, we also have

$$\nu_j^2 = (1-\alpha)^2\nu_{j-1}^2 + \left(\frac{2c_1\alpha}{\sqrt{\ell}}\right)^2$$
$$\leq (1-\alpha)\nu_{j-1}^2 + \alpha\nu_{j-1}^2 = \nu_{j-1}^2.$$

The inequality $\left(\frac{2c_1\alpha}{\sqrt{\ell}}\right)^2 \leq \alpha\nu_{j-1}^2$ follows as long as $\ell = c \cdot \frac{\alpha \log(2/\delta)}{\epsilon^2}$ for sufficiently large constant $c$. We have thus proven (11) and the theorem follows. $\qquad\square$

## C  DeltaShift++ Analysis

In this section, we prove Theorem 4.3. Before doing so, we include pseudocode for the DeltaShift++ algorithm. We let $\widehat{h}_\ell^{++}(A)$ denote the output of the Hutch++ algorithm from [30] run with $\ell$ matrix-vector multiplications – we refer the reader to that paper for details of the method.

---
**Algorithm 4** DeltaShift++
---
**Input**: Implicit matrix-vector multiplication access to $A_1, ..., A_m \in \mathbb{R}^{n \times n}$, positive integers $\ell_0, \ell$, damping factor $\gamma \in [0, 1]$.
**Output**: $t_1, \ldots, t_m$ approximating $\operatorname{tr}(A_1), \ldots, \operatorname{tr}(A_m)$.
  Draw $\ell_0$ random $\pm 1$ vectors $g_1, \ldots, g_{\ell_0} \in \mathbb{R}^n$
  Initialize $t_1 \leftarrow h_{\ell_0}^{++}(A_1)$
  **for** $j \leftarrow 2$ **to** $m$ **do**
    $t_j \leftarrow \gamma \cdot h_\ell^{++}(A_j) + (1-\gamma)\left(t_{j-1} + h_\ell^{++}(A_j - A_{j-1})\right)$
  **end for**
---

**Theorem 4.3** (Restated). *For any $\epsilon, \delta, \alpha \in (0,1)$, DeltaShift++ (Algorithm 4) run with $\ell_0 = O(\sqrt{1/\delta}/\epsilon)$, $\ell = O(\sqrt{\alpha/\delta}/\epsilon)$, and $\gamma = \alpha$, solves Problem 2 with*

$$O\left(m \cdot \frac{\sqrt{\alpha/\delta}}{\epsilon} + \frac{\sqrt{1/\delta}}{\epsilon}\right)$$

*total matrix-vector multiplications involving $A_1, \ldots, A_m$.*

*Proof.* DeltaShift++ is based on a slightly different formulation of the recurrence in (6) that was used to design DeltaShift. In particular, rearranging terms, we see that Equation (6) is equivalent to:

$$\mathrm{tr}(A_j) = (1 - \gamma)\left(\mathrm{tr}(A_{j-1}) + \mathrm{tr}(\Delta_j)\right) + \gamma\mathrm{tr}(A_j), \quad \text{where} \ \ \Delta_j = A_j - A_{j-1}. \tag{12}$$

Following this equation, DeltaShift++ approximates $\mathrm{tr}(A_j)$ via:

$$t_j = \gamma h_\ell^{++}(A_j) + (1 - \gamma)\left(t_{j-1} + h_\ell^{++}(A_j - A_{j-1})\right). \tag{13}$$

As in the analysis of DeltaShift, we bound the variance of $t_j$ recursively, showing that it is less than $\delta\epsilon^2$. We start with the base case. From Fact 4.2 and our assumption in Problem 2 that $\|A_1\|_* \leq 1$, we have the $\mathrm{Var}[t_1] \leq \epsilon^2\delta$ as long as $\ell_0 = \frac{4}{\sqrt{\delta}\epsilon}$. Then the recursive case:

$$\begin{aligned}
\mathrm{Var}[t_j] &= \gamma^2\mathrm{Var}[h_\ell^{++}(A_j)] + (1 - \gamma)^2\mathrm{Var}[h_\ell^{++}(\Delta_j)] + (1 - \gamma)^2\mathrm{Var}[t_{j-1}] \\
&\leq \frac{16\gamma^2\|A_j\|_*^2}{\ell^2} + \frac{16(1 - \gamma)^2\alpha^2}{\ell^2} + (1 - \gamma)^2\delta\epsilon^2 \\
&\leq \frac{16\alpha^2}{\ell^2} + \frac{16\alpha^2}{\ell^2} + (1 - \alpha)\delta\epsilon^2 \\
&\leq \frac{\alpha}{2}\delta\epsilon^2 + \frac{\alpha}{2}\delta\epsilon^2 + (1 - \alpha)\delta\epsilon^2 = \delta\epsilon^2,
\end{aligned}$$

where the last inequality holds as long as $\ell = \frac{4\sqrt{2\alpha/\delta}}{\epsilon}$. Given a bound on the variance of $t_1, \ldots, t_m$, we then just apply Chebyshev's inequality to obtain the required guarantee for Problem 2. $\square$

**Choosing $\gamma$ in practice.** As in Section 3.2, we would like to choose $\gamma$ automatically without the knowledge of $\alpha$. We can do so in a similar way as before by minimizing an approximation to the variance of $t_j$ over all possible choices of $\gamma$. This is a bit trickier than it was for DeltaShift because the stated variance of Hutch++ in Fact 4.2 depends on the nuclear norm of the matrix being estimated, which is not easy to approximate using stochastic estimators. However, it turns out that this variance is simply an upper bound provided by the analysis in [30]: the precise variance bound depends on $\|A - PA\|_F^2$ where $P$ is a low-rank projection matrix obtained when running Hutch++. This quantity can be computed via stochastic trace estimation, and by doing so we obtain an expression for a near optimal choice of $\gamma$, exactly as in Section 3.2. This near optimal $\gamma$ is what is used in our experimental evaluation. Note that obtaining this $\gamma$ is what necessitated the reformulated recurrence of (13), as we only require the variance of Hutch++ run on two fixed matrices at each iteration: $A_j$ and $\Delta_j = A_j - A_{j-1}$ DeltaShift on the other hand involved a matrix $\widehat{\Delta}_j = A_j - (1 - \gamma)A_{j-1}$ that depended on $\gamma$. It would not be possible to easily obtain a direct equation for the variance of Hutch++ applied to this matrix as the matrix $P$ computed by Hutch++ would change dependeing on $\gamma$.

Similar to DeltaShift, we choose $\gamma_i$ at each step $i$ that minimizes the variance of estimate at that particular step. Specifically, letting $K_A = \|A - A_k\|_F^2$ where $A_k$ is a rank-$k$ approximation to matrix $A$, and $v_i$ be the variance of estimate at time step $i$, we obtain

$$\gamma_i^* = \min_\gamma \left[\frac{\gamma^2 8K_{A_i}}{\ell} + (1 - \gamma)^2\left(v_{i-1} + \frac{8K_{\Delta_i}}{\ell}\right)\right] = \frac{8K_{\Delta_i} + \ell v_{i-1}}{8K_{A_i} + \ell v_{i-1} + 8K_{\Delta_i}} \tag{14}$$

Note that similar to DeltaShift, we can reuse the matrix-vector products to calculate near-optimal $\gamma_i$ at each step.

# D Experimental details

**Allocation of matrix-vector products for Restart and NoRestart methods:** The rationale behind the Restart method is using higher number of matrix-vector products for the first matrix in the sequence, letting us use less for subsequent matrices, followed by restarts at set intervals. Note that this still lets us take advantage of relatively small perturbations to the matrices. Following this motivation, for a sequence of 100 matrices we restart every $q = 20$ time steps. The $Q$ matrix-vector multiplications (total matrix-vector products) were evenly distributed to each block of 20 matrices, and then $1/3$ of those used for estimating the trace of the first matrix in the block, and the rest split evenly among the remaining 19. For NoRestart, the same number of vectors were allocated to $A_1$ as for Restart, and the rest evenly divided among all 99 remaining steps, which results in better accuracy for $A_1$ compared to Hutchinson's and DeltaShift.

**Synthetic data:** For synthetic data experiments, we consider a random symmetric matrix $A \in \mathbb{R}^{n \times n}$ and random perturbation $\Delta \in \mathbb{R}^{n \times n}$ to $A$ for 100 time steps, with $n = 2000$. We consider two cases, one where the perturbations are small (Fig. 1(a)) and one where the perturbations are significant (Fig. 1(b)). For both cases, $A_1$ (first matrix in the sequence) is a symmetric matrix with uniformly random eigenvectors and eigenvalues in $[-1, 1]$. For small perturbations, each perturbation is a random rank-1 matrix: $\Delta_j = 5e^{-5} \cdot r \cdot gg^T$ where $r$ is random $\pm 1$ and $g \in \mathbb{R}^n$ is random Gaussian. For the large perturbation case, each $\Delta_j$ is a random rank-25 positive semidefinite matrix. As such, $A$'s trace and Frobenius norm monotonically increase over time, which is reflected in increasing absolute error among all algorithms.

**Estimating natural connectivity:** Application of dynamic trace estimation to the problem discussed in [43] involves estimating the natural connectivity of a dynamic graph (which is $\mathrm{tr}(\exp(B))$ for an adjacency matrix $B$). We use Lanczos with 15 iterations to approximate the matrix-vector product $\exp(B) \cdot g$ and start with an accurate estimate for the first matrix in the sequence (using 5000 matrix-vector products with Hutchinson's). For estimating the trace of $\Delta_i$ matrices, we use $\ell = 50$ for DeltaShift and DeltaShift++ across 100 time steps. Note that for estimating trace of matrix $A$, Hutch++ allocates the number of matrix-vector products as $\ell/3$ for three separate purposes (refer [30]). For estimating trace of $\Delta_i$, we can divide these matrix-vector products as $\ell/5$ instead of $\ell/6$ (for two matrices $A_{i+1}$ and $A_i$) as we can reuse one set of matrix-vector products.

**Hessian spectral density:** Approximating the spectral density of Hessian requires computing the trace of polynomials of the Hessian. We consider the Chebyshev polynomials. The three term recurrence relation for the Chebyshev polynomials of first kind is:

$$T_0(H) = I \qquad T_1(H) = H \qquad T_{n+1}(H) = 2HT_n(H) - T_{n-1}(H) \qquad (15)$$

Here $I$ is the identity matrix. As Chebyshev polynomials form orthogonal basis for functions in range $[-1, 1]$, as a first step we estimate the maximum eigenvalue of the Hessian using power iteration and scale $\tilde{H} = H/\lambda_{\max}$. Like trace estimation, power iteration requires computing Hessian-vector products, which we compute approximately using the PyHessian library [48].[2] For a given neural network and loss function, PyHessian efficiently approximates Hessian-vector products by applying Pearlmutter's method to a randomly sampled batch of data points [32]. To compute matrix-vector products with $T_0(\tilde{H}), T_1(\tilde{H}), \ldots, T_q(\tilde{H})$, which are needed to approximate the trace of these matrices, we simply implement the recurrence of (15) using PyHessian's routine for Hessian-vector products. Multiplying by $T_q(\tilde{H})$ requires $q$ Hessian-vector products in total. As computing ground truth values is impossible in this setting, we use Hutchinson's with 500 matrix-vector products as the ground truth values.

**Experimental setup:** All experiments were run on server with 2vCPU @2.2GHz and 27 GB main memory and P100 GPU with 16GB memory.

---

[2]Available under an MIT license.