# OpenReview forum: "Dynamic Trace Estimation"
_NeurIPS.cc/2021/Conference — NeurIPS 2021 Poster_

### Official Review · Reviewer_hb2j · 2021-07-15

**Rating:** 6
**Confidence:** 2

**Summary:**

This work considers the problem of dynamic trace estimation. Specifically, if there is a sequence of matrices $A_1, \dots, A_n$, and the norm of $A_i - A_{i-1}$ is small relative to the norm of $A_i$, we can estimate the trace of each matrix efficiently. The paper gives an efficient algorithm, the convergence analysis and conducts relavant experiments.


**Limitations And Societal Impact:**

This method seems very specific to this problem. I don't know if it can be used in other time varying optimization problems, or would be useful for other machine learning tasks. The trace estimation problem has two special properties: The trace is linear, so we can easily divide it into subproblems; and the estimation error has a relative bound. If we have a time varying linear regression problem instead, it is hard to reasonably assume that the error can be bounded relatively rather than absolutely. Furthermore, although the problem is "dynamic", we are given the matrices $A_i$ and has the freedom to perform the measurement on the linear combinations of $A_i$ as we want. This is very different from most time series datasets where we can only observe data, not measure it.

I not satisfied with Section 3.2, the theoretical justification of this adaptive step is lacking. This can be a main focus of the paper.

One immediate application of this dynamic trace estimation I can think of, is in the stepsize for gradient descent (quasi-Newton method). At each step, we can use the trace of the inverse Hessian (divided by dimension) as the stepsize. Then the difference of two Hessians is diminishing as the iterates get closer to the optimizer. If the difference is vanishing, how can we choose the damping parameter? And I guess the answer to this can give us a very good quasi-Newton method. It is also related to the previous problem. We must choose the damping parameter adaptively.

**Main Review:**

The paper is well-organized and I don't see any major flaws in the proofs. The idea of damped variance reduction is quite good. But I worry that setting of this paper is too specific and is not very important for other types of problems.

**Time Spent Reviewing:**

6

---

> ### Author Response · Authors · 2021-08-09
> **Authors response**
>
> We thank the reviewer for pointing out an interesting potential application for DeltaShift: quasi-newton methods that utilize the trace of the inverse of the Hessian. This is a domain where we believe DeltaShift can definitely help speedup the optimization method.
>
> Regarding adaptively choosing the damping parameter $\gamma$: The method in Section 3.2 is a greedy heuristic -- at each step, we choose $\gamma$ to minimize an estimate for the variance at each step. This approach is motivated  by the observation that the difference in norm between matrices, $\alpha$, is typically unknown in advance and might even change over time. We feel that the greedy approach is intuitive, and the method performs very well experimentally. However, we agree that additional theoretical exploration of the approach could be interesting for future work. For example, a key advantage of the method in Section 3.2 is that it naturally reduces the damping factor as differences between the $A$ matrices decrease, which is important for e.g. optimization applications, as the reviewer points out. While beyond the scope of the current paper, one natural theoretical question is if we can prove that the adaptive gamma method uses a near-optimal number of matrix vector multiplies -- specifically $(\sum_{i=1}^m \alpha_i) /\epsilon^2$ --  for any sequence of matrices $A_1, …, A_m$ where $||A_i - A_{i+1}||_F \leq \alpha_i$.

---

> > ### Comment · Reviewer_hb2j · 2021-08-18
> > **Reply to the authors**
> >
> > Thanks for the detailed explanation! After reading all the comments, I am happy to increase my score to 6.

---

> > > ### Author Response · Authors · 2021-08-28
> > > **Re:**
> > >
> > > Great -- thank you. Just a heads up: we noticed that your review score still seems to be at a 5 instead of the 6. Although maybe we don't see these updated from the authors side?

---

### Official Review · Reviewer_jK9X · 2021-07-16

**Rating:** 6
**Confidence:** 4

**Summary:**

The authors proposed a method for estimating the trace for a sequence of matrices that only changes for a small amount at each step. The method originates from the well-known stochastic trace estimator, but bring it to a new practical setting (e.g. Hessian at learning iterations, evolving network, etc). By an interesting treatment of the iterative process (i.e. damping), the resulting algorithm could cut the computational cost by a factor of the size of the tolerable error. In addition, the authors included the recent development/improvement on the original Hutchinson estimator, to get better versions of their algorithm under more assumptions.

**Ethical Concerns:**

There is no ethical concerns from the content of this paper.

**Limitations And Societal Impact:**

There is no negative societal impact of this work.

**Main Review:**

Please address this and correct me if I'm wrong, in the inequality between line 175 and 176, $(1-\alpha)^2\delta\epsilon^2+4\alpha/\ell\to\frac{3}{2}\delta\epsilon^2$ as $\alpha\to 0$. Thus the inequality doesn't seem to hold here.

Originality: I think the work is novel and this dynamic setting is probably never looked at before. The work builds on top of the well-known Hutchinson estimator, but the main contribution has great ideas that are totally independent from previous work. The paper also has good reference and sourced all the relevant work.

Quality: The paper is technically sound for the most part, but please respond about my question at the top. Nonetheless the experiment results are promising and shows the algorithm works well in practice.

Clarity: The paper is well-written and easy to follow. The list of reference is good. One thing bothers me a little is that the authors claimed in multiple places that $tr(\Delta_j)$ is easier to estimate and we can we can approximate $tr(\Delta_j)$ more accurately than $tr(A_j)$. While I get authors' point in the context of this problem, the statement on its own isn't very accurate because we usually look at relative error in this kind of estimations, and $tr(\Delta_j)$ isn't obviously easier in that sense.

Significance: I think the work is important and have many applications as suggested by authors' experiments. While I'm not sure about how much scalability problem we have in practice given the efficiency of the original estimator, the proposed work makes clear improvements.

Other comment: While damping seems a great idea in general, the authors also mentioned that sometimes $\Delta_j$ can have special properties (e.g. low-rank, diagonally concentrated after reordering, etc). In this case, the damping would destroy those structures.


**Time Spent Reviewing:**

2

---

> ### Author Response · Authors · 2021-08-09
> **Authors response**
>
> Thank you for acknowledging the novelty of our idea and clarity of the paper. We address the comments raised below.
> 1. Regarding the inequality between line 175 and 176: the statement overall is correct, but there is a typo in Line 174 that we accidentally carried into the derivation. Since $\hat{\Delta}_j$ has Frobenius norm $\leq 2\alpha$, by Fact 2.1, the variance of estimator for tr$(\hat{\Delta}_j)$ should be $8\alpha^2/l$, not $4\alpha/l$ as currently stated. Substituting with $\ell = 8\alpha/(\epsilon^2 \delta)$ gives $(1-\alpha)^2\delta \epsilon^2 + 8\alpha^2/l =  (1-\alpha)^2\delta \epsilon^2 + \alpha\delta \epsilon^2  \leq  \delta \epsilon^2$, as required. Thank you for catching this, and we will fix it for the camera-ready version of the paper.
> 2. The point about tr$(\Delta_j)$ not necessarily being easier to estimate is a good one, and something we will add some discussion about. We are working without any structural assumptions on $A_1, …, A_m$ (specifically, we don’t assume these matrices are positive semi-definite), in which case relative error guarantees are difficult to obtain. For additive error guarantees, the variance of Hutchinson’s estimator depends on the squared frobenius norm of the matrix, which we do expect to be smaller for the $\Delta$ matrices. However, if $A_1,...., A_m$ have structure -- e.g. they are all PSD -- then we agree that the advantage of estimating traces of the $\Delta$ matrices is less clear. Experimentally, we did still find that DeltaShift offered an improvement over Hutchinson’s (see discussion of natural connectivity experiment, Line 296), but it’s difficult to establish a clear theoretical advantage.
> 3. We agree with the reviewer that introducing the damping factor $\gamma$ (Equation 6) can destroy structure in the $\Delta$ matrices, which can be problematic. This is an important observation, and actually a key issue we address with the alternative DeltaShift++ algorithm (Appendix C, Algorithm 4). This algorithm uses a reformulation of Equation 6 (Equation 12) which let’s us keep the damping factor $\gamma$, but involves a direct estimate for $\Delta_j = A_{j} - A_{j-1}$. This let’s us exploit matrix structure. For example, the recently proposed Hutch++ works better for PSD matrices and low-rank matrices. We show experimentally that when the $\Delta’s$ are approximately low-rank, DeltaShift++ clearly improves on the standard DeltaShift method (Fig 4).

---

> > ### Comment · Reviewer_jK9X · 2021-08-30
> > **Reply to authors**
> >
> > Thank you for addressing my concerns, and please fix the error in the camera-ready. I don't think raising my score will make a big difference here, thus I would like to maintain my score of 6.

---

### Official Review · Reviewer_94L4 · 2021-07-17

**Rating:** 7
**Confidence:** 4

**Summary:**

This paper studies dynamic setting of trace estimation problem; compute the trace of a sequence of matrices efficiently. The authors consider the differences of consecutive matrices are small and utilize the linearity of trace and Hutchinson estimator, which returns trace values with only matrix-vector multiplications. In addition, in order to reduce the variance of trace estimations, they adopt a damping strategy; it controls the scale of the input matrix to estimate and reduces its variance appropriately. They propose a closed form of the optimal damping factor. Experimental results show the superiority of the proposed algorithm with various problems of trace estimation under both synthetic and real-world datasets.

**Limitations And Societal Impact:**

The paper is quite well-written and easy to understand. But, it would be good to highlight concrete applications that I provide in the experiment section. Also, I suggest providing some details of technical terms (e.g., Hanson-Wright inequality in line 127) for non-expert readers.

**Main Review:**

This work proposes an efficient method for the dynamic trace estimation problem by utilizing well-known Hutchinson trace estimation and variance reduction technique (aka control variate). The problem is quite fundamental so that it can be applied to various of practical tasks, e.g., triangle counting, Hessian spectral density estimation and so on. The idea of Hutchinson estimator and variance reduction is not new and widely used in many problems involving computational issues, but the authors analyze with concrete theoretical results.

I think that dynamic trace estimation is somewhat related to greedy algorithm of the determinant maximization problem (aka MAP inference of Determinantal Point Processes). Indeed, [1] tried to dynamically estimate log-determinant (polynomial version of Hutchinson estimator) and analyzed the variance of two distinct Hutchinson estimations when randomness is shared (see Theorem 2 in [1]). More generally, it seems that the proposed method can combine with a polynomial to extend to more various essential problems.

[1] Han, Insu, et al. "Faster greedy MAP inference for determinantal point processes." ICML, 2017.


**Time Spent Reviewing:**

4

---

> ### Author Response · Authors · 2021-08-09
> **Authors response**
>
> Thank you for acknowledging the applicability of the problem and clarity of the paper. We address the comments raised below.
> 1. We agree with the point that our algorithm can be used as a subroutine in the greedy algorithm of determinant maximization from [1] and we thank the reviewer for pointing out the reference. The log determinant algorithm uses a polynomial expansion to approximate matrix vector multiplications with log(A) for a matrix A, which is very similar to our experiment setup for estimating natural connectivity, where the goal is to approximate tr(exp(A)). We instead use the Lanczos algorithm, which is implicitly computing a near-optimal polynomial approximation to exp(A). This seems like an interesting additional and impactful application that we can add experiments for in our camera-ready version. The idea behind Theorem 2 in that paper is also similar to our variance reduction approach, so we will add some discussion on that.
> 2. We can include some more details about the Hanson-Wright inequality and will keep an eye out for other places where the paper can be made more readable for non-experts.
>
> [1] Han, Insu, et al. "Faster greedy MAP inference for determinantal point processes." ICML, 2017.

---

> > ### Comment · Reviewer_94L4 · 2021-08-18
> > **Follow up**
> >
> > Thank you for the response to the comments. I have read other reviews and author response and it is satisfactory. I believe this work leads to interesting implications in camera-ready or future work. Therefore, I maintain my review score.

---

### Official Review · Reviewer_uAQs · 2021-07-23

**Rating:** 7
**Confidence:** 4

**Summary:**

The authors in this paper study the problem of estimating the trace of a matrix that is changing dynamically over several iterations. Under the assumption that the diagonal entries of the matrix are not directly accessible, but multiplication with a vector $Av$ (or the quadratic form $v^{\top}Av$), the Hutchinston estimator is the baseline algorithm for comparison. The Hutchinston estimator takes the average of $\epsilon^{-2} \log 1/\delta,$ estimates $g_i^{\top} A g_i$ where $g_i$ is a random vector with iid gaussian variables, or Rademacher ($\pm 1$) variables, and is correct with probability $1-\delta.$

The model is that we have a sequence of matrices $A_1, \ldots, A_m$ each with $\|| A_i \|| \le 1$ and $\||A_{i+1} - A_{i} \|| \le \alpha,$ (where the matrix norm used is the Frobenius norm) and we wish to output estimates of $tr(A_1),\ldots,tr(A_m)$ such that each estimate is accurate to $\pm \epsilon.$ Using a Hutchinston estimator at each iteration will require $m \epsilon^{-2} \log m/\delta$  where all the answers are accurate with probability $1-\delta.$

The paper presents an algorithm that is able to maintain accurate estimates for such a dynamically changing $A$ using only $(m \alpha + 1 )\epsilon^{-2} \log m/\delta,$ which is essentially better by a factor of $\alpha.$ The authors also show a lower bound claiming to show that this method cannot be improved unless the original Hutchinston method is improved.

The algorithms extend their algorithm to the case where you have a stronger nuclear norm bound assumption, and present numerical simulations demonstrating their advantages.

**Main Review:**

I think this is a nice algorithm based on a clean and simple idea that could lead to direct gains in some settings, e.g. estimating the trace of the hessian while training a neural network. The key idea is to maintain a trace estimate of a moving (exponentially) weighted average of the matrices, and to estimate the trace of the difference at each step using the Hutchinston estimator.

A few comments:
1. A nicer baseline to compare against is a Hutchinston estimator that is restarted every $1/\alpha$ iterations.
2. The lower bound is weaker than the claim that "this method is the best possible". The lower bound only shows that using the case $\alpha \approx 1/m,$ this problem recovers the original estimation problem for hutchinston estimator. It does not rule out running time bounds such as $m + \sqrt{m \alpha} \epsilon^{-2} \log m/\delta.$
3. I think your algorithm can also work in the weaker model where we only have access to the quadratic forms. I think that's the usual model for the Hutchinston estimator too.

**Time Spent Reviewing:**

3-4

---

> ### Author Response · Authors · 2021-08-09
> **Authors response**
>
> Thank you for acknowledging the simplicity and performance gains obtained by our approach. We address the comments raised below.
> 1. Comparing to a restarted version of Hutchinson’s estimator is a good idea, and as we mention in the paper (line 149), it is actually possible to analyze this approach to obtain a guarantee similar to that of DeltaShift. We call this the “Restart” algorithm and compare it to DeltaShift in Figure 1. While it performs reasonably well (worse than DeltaShift, but better than other methods), a disadvantage is that it is difficult to implement “Restart” in the setting where  $\alpha$ is not known in advance, or changes over time. In Figure 1 we chose the best possible restart frequency to compare against, but for later experiments it was difficult to find a frequency that worked well across the entire set of dynamic updates.
> 2. We agree with the point raised by the reviewer and thank them for pointing it out. In the camera ready version will change how we discuss the lower bound to make it more clear that it only holds for a certain parameter regime ($\alpha = 1/m$). We will also reword Lemma 4.1 accordingly.
> 3. This is correct that our algorithm can work in the quadratic form model, and we will add a mention of this. However, it is only true when using a constant damping factor $\gamma$. The proposed heuristic for adaptively choosing $\gamma$ (Section 3.2) involves $g^TA_{j-1} A_j g$, which requires the matrix-vector model to compute. Finding a way to select a near-optimal $\gamma$ in the quadratic form model is an interesting direction for future research.

---

### Decision · Program_Chairs · 2021-09-27

**Decision:**

Accept (Poster)

**Comment:**

This paper considers estimating the trace of a sequence of matrices $A_1, A_2, … A_t$ under the constraint that $||A_i - A_{i+1}||\_2$ is small. The proposed algorithm outperforms the baseline which repeatedly applies Hutchinson's stochastic trace estimator in terms of sample complexity. This paper receives unanimous support from the reviewers. Thus, I recommend acceptance.